Evolutionary patterns of the SSU rRNA (V4 region) secondary structure in genus Euplotes (Ciliophora, Spirotrichea): insights into cryptic species and primitive traits

Wardani Ratih Kusuma 1
Ahsan Ragib 1 2 3
http://orcid.org/0000-0001-6592-1204 Shin Mann Kyoon 1 mkshin@ulsan.ac.kr
1 Department of Biological Science, University of Ulsan , Ulsan , Republic of South Korea
2 Program in Organismic and Evolutionary Biology, University of Massachusetts Amherst , Amherst, Massachusetts , USA
3 Department of Biological Sciences, Smith College , Northampton, Massachusetts , USA
Gillespie Joseph
Electronic publication date: 2025 Jan 23
Publication date: 2025
Volume: 13
Electronic Location ID: e18852
Received 2024 Apr 16; Accepted 2024 Dec 20
Copyright: © 2025 Wardani et al.
Copyright year: 2025
Copyright holder: Wardani et al.
License: This is an open access article distributed under the terms of the Creative Commons Attribution License, which permits unrestricted use, distribution, reproduction and adaptation in any medium and for any purpose provided that it is properly attributed. For attribution, the original author(s), title, publication source (PeerJ) and either DOI or URL of the article must be cited.
License URL: https://creativecommons.org/licenses/by/4.0/

Keywords: Euplotes, V4 region, SSU rRNA, Secondary structure, Evolutionary pattern, Cryptic species, Primitive character

Funding: Ministry of Education of the Republic of Korea 2021R1I1A2048744 This work was supported by the NRF Korea grant funded by the Ministry of Education of the Republic of Korea (2021R1I1A2048744). The funders had no role in study design, data collection and analysis, decision to publish, or preparation of the manuscript.

==============================
The genus Euplotes, a group of ciliated protists, has attracted attention as a model organism due to its widespread distribution and ease of cultivation. This study examines the evolutionary patterns of the SSU rRNA secondary structure within this genus, aiming to elucidate its role in supporting evolutionary relationships and uncovering cryptic species. By predicting the secondary structure of SSU rRNA and applying the CBC (Compensatory Base Change) concept analysis, we examined 69 species of the genus Euplotes, with 57 SSU rRNA gene sequences retrieved from GenBank and 12 newly sequenced specimens from South Korea. Our analysis revealed significant variations in the V4 region secondary structure, particularly in helix E23_8, across different clades of Euplotes. Reconstruction of the ancestral state indicated a transition from a simpler (Type I) to a more complex (Type II) secondary structure, with several species showing a reversal to Type I especially species in clade VI, suggesting of reverse evolution. In addition, our study identified cryptic species within Euplotes based on differences in the secondary structure of the V4 region, particularly evident in clade VI, where CBC analysis highlighted differences in E. minuta compared to E. vannus and E. crassus. These results highlight the utility of molecular data in refining species boundaries and evolutionary patterns within the genus Euplotes.

Introduction

Genus Euplotes is a remarkably diverse genus of ciliates, comprising more than 100 species (Bisby et al., 2010). This genus is notable for its cosmopolitan distribution and ease of cultivation in the laboratory, as it is commonly found in a wide variety of environment. Euplotes serves as a valuable model organism alongside Paramecium and Tetrahymena (Aury et al., 2006; Greider & Blackburn, 1989; Kruger et al., 1982; Sonneborn, 1975). Euplotes has been extensively utilized in studies investigate the adaptation of single-celled organisms to extreme conditions such as cold environments (La Terza et al., 2001; Marziale et al., 2008; Mozzicafreddo et al., 2021), as well as research in mating on processes and pheromone studies (Di Giuseppe et al., 2011; Gong et al., 2022; Pedrini et al., 2022), the symbiotic relationships between symbionts and hosts (Boscaro et al., 2019a, 2019b; Vannini et al., 2017), and the geographical distribution of unicellular organisms (Syberg-Olsen et al., 2016). Given the importance of Euplotes’s as a model organism, numerous studies have focused on species delimitation and the exploration of new species across diverse environments and biogeographies (Abraham et al., 2021; Han, Pan & Jiang, 2022; Lian et al., 2020; Syberg-Olsen et al., 2016; Valbonesi et al., 2021; Zhao et al., 2018). In the process of delimitation within the genus Euplotes several cryptic species emerge due to their indistinguishable morphological character (Wang et al., 2021; Zhao et al., 2018). Molecular data have mainly played a supporting role in the phylogenetic studies of Euplotes and in the attempt to reveal cryptic species within genus Euplotes (Zhao et al., 2018). However, it should be noted that molecular data can play a more significant role than previously discussed. For example, in the study of the genus Euplotes, molecular data, such as SSU rRNA secondary structure, can serve as a key feature to identify new species (Abraham et al., 2021).

The SSU rRNA secondary structure, which serving as a molecular character, exhibits specific evolutionary patterns that have proven valuable in phylogenetic studies of different taxa (Marin et al., 2003; Rusin et al., 2001; Telford, Wise & Gowri-Shankar, 2005; Voronova & Chelomina, 2020). This is evident in phylogenetic studies of ciliates (Du, Zhao & Tang, 2018; Gao, Katz & Song, 2013; Gao et al., 2014; Li et al., 2008; Wang et al., 2015; Zhang et al., 2014; Zhang et al., 2015). Components of the RNA secondary structure, including stems, inner loops, hairpin loops, and bulges, are features that can support traditional cladistics and contribute to our understanding of the universal tree of life by examine the evolutionary patterns inherent in these molecular characters (Caetano-Anollés, 2002).

The secondary structure of SSU rRNA contains nine variable regions (V1–V9) that hold phylogenetic information. Among these, the V4 region is one of the most studied in ciliates due to its superior ability to resolve phylogenetic relationships compared to the V9 region (Dunthorn et al., 2012, 2014; Santoferrara, McManus & Alder, 2013). The V4 region has been widely used to differentiate closely related ciliate taxa, such as scuticociliates (Gao et al., 2014), litostomes (Strüder-Kypke et al., 2006), and haptorians (Zhang, Simpson & Song, 2012). Furthermore, the V4 region is valuable for estimating the diversity of different protist taxa due to its high mutation rate (Stoeck et al., 2010, 2014; Santoferrara, McManus & Alder, 2013; Dunthorn et al., 2014).

In addition to elucidating taxonomic relationships and evolutionary history, the secondary structure of SSU rRNA can also be used to distinguish between different species. This is achieved through compensatory base change (CBC) analysis on the helices of the secondary structure, which detects nucleotide base changes on both sides of the paired bases. An experimental study by Coleman & Vacquier (2002) demonstrated a correlation between CBCs and sexual compatibility between species. The study found that taxa differing by CBCs, even by just a single CBC in conserved pairing positions, showed differences in sexual compatibility.

Therefore, this study aims (1) to observe the evolutionary pattern of the SSU rRNA secondary structure of V4 region in the genus Euplotes as a speciose and model organism group and its usability in supporting the phylogenetic and evolutionary of the genus Euplotes, and (2) to reveal cryptic species within the genus Euplotes through CBC analysis of the secondary structure of the V4 region.

Materials and Methods

Sample collection and morphological study

Twelve species of Euplotes were collected from various locations in South Korea (Table 1). These specimens were cultured in Petri dishes containing water from their respective habitats. The cells were initially observed in their live state using a stereomicroscope (Nikon SMZ800, Nishioi, Shinagawa-ku, Tokyo) to assess their typical shape, movement, and behavior. For more detailed analysis, a differential interference contrast (DIC) microscope (Axio Imager A.1, Carl Zeiss, Oberkochen, Germany) was used, allowing for magnification between 100× and 1,000× for both live and stained samples.

Table 1 Characterization of SSU rRNA gene sequences of Euplotes species from Korea.

Taxon	Collection site	SSU rRNA gene	Clade	
GenBank
entry	Length
(nt)	GC
(%)	
Euplotes cf. inkystans	Dokdo, Korea	PP648189	1,794	45.2	V	
Euplotes cf. mutabilis	Dokdo, Korea	PP648190	1,836	44	VI	
Euplotes crenosus	Dokdo, Korea	PP648191	1,972	42	VI	
Euplotes neopolitanus	Dokdo, Korea	PP648192	1,874	42	IV	
Euplotes trisulcatus	Busan, Korea	PP648193	1,750	45.3	V	
Euplotes gracilis	Ulsan, Korea	PP648194	1,803	44	V	
Euplotes muscorum oligomembrana n.subsp.	Ulsan, Korea	PP648195	1,814	43	V	
Euplotes paramuscicola n.sp.	Ulsan, Korea	PP648196	1,807	46	V	
Euplotes vannus pop.1	Pohang, Korea	PP648197	1,840	43	VI	
Euplotes vannus pop.2	Pohang, Korea	PP648198	1,846	43	VI	
Euplotes n. sp.	Dokdo, Korea	PP648199	1,807	45	V	
Euplotes sp.	Dokdo, Korea	PP648200	1,797	45	V	

DNA extraction, amplification, and sequencing

Genomic DNA extraction was performed using the RED Extract-N-Amp Tissue PCR Kit from Sigma (St. Louis, MO, USA), according to the manufacturer’s instructions. For polymerase chain reaction (PCR), the forward primer EUK A (5′-GAC CGT CCT AGT TGG TC-3′) and the reverse primer EUK B (5′-CTT GGA CGY CTT CCT AGT-3′) were used, as described by Medlin et al. (1988). PCR amplification was performed using the TaKaRa ExTaq DNA polymerase kit from TaKaRa Bio-medicals (Otsu, Japan) according to this specific protocol: an initial denaturation at 94 °C for 2 min, followed by 37 cycles of denaturation at 95 °C for 30 s, annealing at 50 °C for 40 s, and extension at 72 °C for 4 min. This was followed by a final extension at 72 °C for 10 min (Kim et al., 2011). Sequencing was performed with bidirectional sequencing using the primers used in the PCR reaction (EUK A and EUK B).

Predicting the secondary structure of the V4 region of SSU-rRNA and CBC (compensatory base change) concept analysis

To predict the secondary structure of the SSU-rRNA, we performed an alignment of 69 SSU rRNA gene sequences from the species of genus Euplotes retrieved from GenBank (NCBI) and together with newly sequenced species from Korea (Table 1), related genera members (Certesia quadrinucleata, Aspidisca fusca, Euplotidium Rosati, Diophrys scutum, Uronychia xinjiangensis, and Discocephalus pararotatorius) selected as an outgroup. The alignment was performed using SSU-ALIGN and based on the alignment results from SSU-ALIGN, it was determined that the V4 region is a longer variable region (163–261 bp) (Table S1) and has more mutations compared to other variable regions (Fig. S1). From the SSU-ALIGN results, we isolated the V4 region of the SSU rRNA gene (Nawrocki, 2009). To generate consensus secondary structure of the V4 region of genus Euplotes member, we used RNAalifold software (Hofacker, 2008), the consensus secondary structure of V4 region used as a reference for predicting the secondary structure of the V4 region for each species. The prediction of the secondary structure for each species was achieved by using MFOLD, which calculates the minimum energy (Zuker, 2003). To guide the construction of the secondary structure using MFOLD, we used the consensus secondary structure and several criteria. First, we closed bilateral bulges (internal loops) present in published models if they could consistently form G:C pairs with non-canonical pairing bases in the stem. Second, we did not retain paired structures if multiple non-canonical base pairings occurred, instead of canonical (G:C or A:U) or wobble (G:U) base pairs. Final, in cases where multiple structures were predicted, we selected the structure with either the minimum free energy or the best compatibility with similar sequences (Řeháková et al., 2014; Voigt, Erpenbeck & Wörheide, 2008). The final secondary structure of the hypervariable region of SSU-rRNA was visualized using RNAviz software (De Rijk, Wuyts & De Wachter, 2003).

For the compensatory base change (CBC) analysis, we used 4SALE (Seibel et al., 2006) to detect CBCs between sequence-structure pairs within the alignment. The CBC analysis was applied to members of clade VI within the genus Euplotes (E. minuta, E. cf. mutabilis, E. crenosus, E. japonicum, E. cristatus, E. crassus, and E. vannus) (Fig. S2).

Reconstruction of ancestral state

For the ancestral state analysis of the V4 (SSU-rRNA) secondary structure, we used representative species from the genus Euplotes and related genera (Certesia quadrinucleata, Aspidisca fusca, Euplotidium rosati, Diophrys scutum, Uronychia xinjiangensis, and Discocephalus pararotatorius) (Table S1). The presence or absence of two additional helices on the extension helix E23_8 (Type I vs. Type II) (Fig. 1) was mapped onto the best-likelihood tree generated by RAxML analysis on the CIPRES platform (Miller, Pfeiffer & Schwartz, 2011). The character matrix and subsequent ancestral state reconstruction were performed using the parsimony model in Mesquite software (version 3.70) (Maddison & Maddison, 2021) (Table S1).

Figure 1 Summary of V4 secondary structure on Euplotes and related genera.

(A) Helices of V4 secondary structure marked in red, representing helices commonly found in Prokaryotes (Bacteria), and in green mark, representing extended helices in Eukaryotes (Lee & Gutell, 2012), (B) V4 secondary structure type I in genus Euplotes members, featuring one hairpin loop at the end of E23_8, (C) V4 secondary structure type II in genus Euplotes members, with two extra helices (E23_11 & E23_12).

Results

Significant characters of SSU rRNA secondary structure

In this study, we focused on the secondary structures of the V4 region. This choice was prompted by the observation of a significant number of mutations in this region, as indicated by the results of SSU-Align and primary sequence alignment. The consensus secondary structure predicted for the V4 region by the RNAalifold software identified four major helices (Fig. 1). In general, the secondary structure of each member of the genus Euplotes consists of four helices, with variations observed specifically in helix E23_8. These variations include two additional helices or hairpin loops at the end of the helix, highlighting the structural diversity within the genus Euplotes in the V4 region (Fig. 1).

Each clade depicted in the phylogenetic tree (Fig. S2) has a distinct secondary structure pattern. Members of clade I display a single large hairpin loop composed of 21 nucleotides in helix E23_8 (Fig. 2). This structural feature is also shared by the members of clade II, but smaller in size, consisting of 13 nucleotides (Fig. 2). Clade III is divided into two groups based on the characteristics of the V4 secondary structure. The first group include species with a hairpin loop consisting of five nucleotides (Fig. 2). Members of this group include E. curdsi, E. dominicanus, E. estuarianus, E. nobili, E. raikovi, and E. shii (Fig. 3). The second group is characterized by a long helix in E23_8 and the presence of two additional helices (E23_11 & E23_12) (Fig. 2). This group includes E. bergeri, E. elegans, E. qatarensis, and E. wuhanensis (Fig. 3). All members of clade IV share a V4 secondary structure characterized by the presence of two additional helices (E23_11 & E23_12), with the helix E23_8 shorter compared to the second group of clade III (Fig. 2). All members of clade V have two additional helices, except for three species that have a single hairpin loop at the end of helix E23_8 (Fig. 3). These three species are E. trisulcatus (hairpin loop consisting of 32 nucleotides), E. trisulcatus (hairpin consisting of 17 nucleotides) and E. shini (hairpin loop consisting of 29 nucleotides) (Fig. 2). In clade VI, members share a common pattern of a single large hairpin loop composed of 25–29 nucleotides. Interestingly, E. minuta, a member of clade VI, deviates from this pattern and show two additional helices (E23_11 & E23_12) (Fig. 3 and Fig. S2).

Figure 2 Several species of the genus Euplotes and their V4 secondary structure (E23_8, E23_11, and E23_12).

Figure 3 Ancestral state analysis of V4 in the genus Euplotes, with V4 secondary structure “Type I” and “Type II” as characters.

In addition to secondary structure, we applied compensatory base change (CBC)analysis to the members of clade VI, focusing specifically on the genus Euplotes vannus-minuta-crassus complex. The CBC shows that E. minuta has one CBC compared to other members of clade VI (Fig. 4).

Figure 4 Euplotes minuta shows one compensatory base change (CBC) compared to another member of Clade VI.

Ancestral state of V4 secondary structure within genus Euplotes

The ancestral state was analyzed using the V4 secondary structure within the genus Euplotes to discern the evolutionary pattern of molecular characters, with particular focus on helix E23_8 (hairpin loop vs. two additional helices) (Fig. 1). The V4 secondary structure type I represents a molecular feature inherited from the ancestor of the genus Euplotes, and it is retained in members of clade I and II (Fig. 3). The ancestral state analysis shows that Type II of the V4 secondary structure, indicating the addition of two helices, is evolve feature compared to Type I of the V4 secondary structure (Fig. 3). Type II V4 secondary structure is commonly observed in almost all members of clade III to V with the exceptions of a few species that have Type I of the V4 secondary structure as their molecular character. These species include E. dominicanus, E. estuarinus, E. curdsi, E. nobilii, and E. shii in clade III, and E. trisulcatus, E. bisulcatus and E. shini in clade V (Fig. 3).

An interesting observation arises in the members of clade VI, where all members except E. minuta have Type I of the V4 secondary structure as their molecular character. Clade VI appears to represent a relatively recent clade in the phylogenetic tree of the genus, and these results suggest a remark reappearance of primitive characters reappearing in the most advanced species within this clade (Fig. 3).

Discussion

Reverse evolutionary pattern of the V4 secondary structure within the genus Euplotes

The Type I of V4 secondary structure is regarded as simpler compared to Type II, as it shares structural similarities to the V4 secondary structure found in prokaryotes. In prokaryotes, the E23 (V4) region typically lacks an additional helix, indicating a primitive feature compared to eukaryotes (Lee & Gutell, 2012). This primitive feature is observed in members of the genus Euplotes and outgroup genera used as the earliest divergence group at the base of the phylogenetic tree, as supported by molecular clock analysis (Fig. S3). The primitive character of the V4 secondary structure was also observed in some later diverging species, particularly in Group VI (Fig. 3 & Fig. S1).

The secondary structure of V4 in the common ancestor of the genus Euplotes is characterized by Type I, which later evolved into the more complex Type II structure (Fig. 3). Interestingly, in certain evolved species there was a reduction in the E23_11 and E23_12 helices, resulting in a reversion to the Type I V4 secondary structure (Fig. 3). This pattern is suggestive of reverse evolution, where a character state changes to resemble the ancestral state, involving reversals and regressions that reflect evolutionary patterns of reversion to earlier or simplified forms after initially becoming more complex (Porter & Crandall, 2003; Teotónio & Rose, 2001).

The structural variations observed in the V4 secondary structure result from deletions or insertions in the V4 SSU rRNA region (Fig. S4). This pattern suggests that the occurrence of insertions or deletions in this region implies its lack of conservation and limited relevance to ribosome function, and thus the high degree of evolutionary change in this region is unlikely to have a significant impact on ribosomal function (Wuyts, Van de Peer & De Wachter, 2001). Although this region may not directly affect ribosome function, it is important because of the exceptionally high mutation rate within the SSU rRNA gene sequence. This region is likely to play a critical role in maintaining the free energy level to support the conservation of the SSU rRNA secondary structure. This study shows that the Type II of the V4 secondary structure has a more favorable free energy profile when compared to Type I (Table S1). The reverse evolution of the V4 secondary structure is not driven by the energy favorability, but is likely due to deletions that occur within this specific region.

The evolutionary pattern of V4 secondary structure supports the primitive nature of the basal clade in the genus Euplotes

The genus Euplotes shows distinct groupings, with a basal group (clade I) containing species such as E. huizhouensis, E. petzi, and E. sinicus, and subsequent divergent groups (clades II to VI) (Fig. 3). The evolutionary pattern of the V4 secondary structure shows that clade I, as a basal group, has primitive or ancestral characteristics compared to the later evolved groups. This pattern extends beyond molecular characters, morphological characters also follow a similar trend. Specifically, the basal group (clade I) displays a distinctive double-pattern argyrome character, representing the ancestral state of the genus Euplotes. In addition, these species display 10 fronto-ventral cirri (FVC), another character considered primitive or plesiomorphic in the genus Euplotes. In summary, molecular evidence from the V4 secondary structure supports the idea that species in the basal group (clade I) of the genus Euplotes are characterized by primitive traits compared to species in later diverging groups (clades II to VI), as reflected in their morphological characters (Syberg-Olsen et al., 2016; Zhao et al., 2018).

Cryptic species within the genus Euplotes in clade VI are revealed by the CBC concept in the V4 secondary structure

In this study, the CBC concept is applied to the members within clade VI, that potentially contain cryptic species. The presence of cryptic species makes it difficult to distinguish the species by morphological characters. By examining the secondary structure of V4, it is clear that E. minuta has a different structure compared to other members of clade VI. The CBC shows changes in nucleotide bonds in E. minuta compared to E. crassus, E. vannus, E. cf. mutabilis, and E. japonicum. To date, cryptic species have been identified among E. minuta, E. crassus, and E. vannus, which can be distinguished mainly by cell size alone (Dini, 1984; Valbonesi, Ortenzi & Luporini, 1988). However, the presence of CBC in E. minuta allows us to reasonably conclude that E. minuta represents a separate species from E. crassus and E. vannus. Furthermore, the CBC concept in E. crassus and E. vannus is consistent with evidence that both species are capable of mating under laboratory conditions (Valbonesi, Ortenzi & Luporini, 1988). This mating compatibility results from their morphological and chemical compatibility, in particular their pheromones. Considering both mating compatibility and the CBC concept, it is plausible that these two species are more appropriately classified as a single species.

Conclusions

The study of the V4 secondary structure within the genus Euplotes provides valuable insights into the evolutionary dynamics of this group. The observed pattern of reverse evolution, in which the V4 structure reverts from the more complex Type II to the simpler Type I, suggests a reversion to ancestral features. Furthermore, the application of CBC analysis within Clade VI reveals the presence of cryptic species, providing a more nuanced understanding of species differentiation within Euplotes. The CBC analysis not only supports the distinct classification of species such as E. minuta but also raises the possibility that E. crassus and E. vannus may share such close genetic similarities that they could potentially be considered as a single species. In conclusion, this research highlights the evolutionary complexity within the genus Euplotes and demonstrates the effectiveness of molecular tools such as V4 secondary structure analysis and CBC in elucidating species relationships and evolutionary history. These findings contribute to a deeper understanding of the processes driving diversity within the genus.

Supplemental Information

Supplemental Information 1 Secondary structure prediction results from the V4 region of SSU rRNA by mFold software.

Supplemental Information 2 Euplotes species from database and their publication based on SSU rDNA sequences.

Supplemental Information 3 Newly sequenced species from Korea.

Supplemental Information 4 SSU-Align software analysis of Euplotes neapolitanus as a representative species, illustrating variable regions (V1–V9) within the SSU secondary structure.

The color coding represents alignment confidence values (low to high: dark orange, orange, yellow, green, cyan, blue), with gaps indicated in gray. Lower confidence values correspond to nucleotide base changes at specific positions, while gaps indicate potential deletions.

Supplemental Information 5 Phylogenetic tree of genus Euplotes based on Bayesian Inference and Maximum Likelihood analysis of SSU rRNA gene.

Node values represent statistical support from BI/ML tree, while node value under 50% and unsynchronized branches are represented with “- “. Newly sequenced species are marked in green.

Supplemental Information 6 Maximum credibility tree showing posterior means of divergence times within the genus Euplotes using the Bayesian dating in BEAST.

The 95% credibility intervals are indicated by bars in each node, and horizontal lines show the dating in million years. The parameter shown in Article S1.

Supplemental Information 7 Alignment of highly mutation regions in V4 of SSU rRNA gene.

Supplemental Information 8 Supplementary materials and methods.

Supplemental Information 9 Secondary structure of Certesia quadrinucleata by Mfold software.

Supplemental Information 10 Secondary structure of Euplotes huizhouensis by Mfold software.

Supplemental Information 11 Secondary structure of Euplotes parabalteatus by Mfold software.

Supplemental Information 12 Secondary structure of Euplotes amieti by Mfold software.

We thank MD Abu Taher (University of Ulsan, University of British Columbia) for providing sample (Euplotes gracilis, Euplotes vannus pop1 and pop2) for analysis in this article.

Additional Information and Declarations

Competing Interests

Author Contributions

DNA Deposition

Data Availability

The authors declare that they have no competing interests.

Ratih Kusuma Wardani conceived and designed the experiments, performed the experiments, analyzed the data, prepared figures and/or tables, authored or reviewed drafts of the article, and approved the final draft.

Ragib Ahsan performed the experiments, authored or reviewed drafts of the article, and approved the final draft.

Mann Kyoon Shin conceived and designed the experiments, authored or reviewed drafts of the article, and approved the final draft.

The following information was supplied regarding the deposition of DNA sequences:

The newly sequenced species are available at GenBank: PP648189 to PP648201.

The following information was supplied regarding data availability:

The data is available in the Supplemental Files.

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
