# Peer review of "Evolutionary patterns of the SSU rRNA (V4 region) secondary structure in genus Euplotes (Ciliophora, Spirotrichea): insights into cryptic species and primitive traits"

_PeerJ, doi:10.7717/peerj.18852_

## Round 0.1 · original submission · Major Revisions

Dear Dr. Wardani and colleagues:

Thanks for submitting your manuscript to PeerJ. I have now received two independent reviews of your work (I was waiting on a third that never came, my apologies), and as you will see, the reviewers raised some concerns about the manuscript. Despite this, the reviewers are optimistic about your work and the potential impact it will have on research studying Euplotes evolution and speciation. Thus, I encourage you to revise your manuscript, accordingly, considering all the concerns raised by both reviewers.

I agree with many of the concerns of the reviewers, and thus feel that their suggestions should be adequately addressed before moving forward. Please understand the limitations of your molecular clock analysis and how the overall findings of your study need to be made clearer in the context of prior work on this genus.

I look forward to seeing your revision, and thanks again for submitting your work to PeerJ.

Good luck with your revision,

-joe

Reviewer 1 ·

Basic reporting

In this study, the authors present phylogenetic analyses and reconstruction of ancestral states using the secondary structure of the 18S rRNA subunit, specifically the V4 region, as a character. The V4 region of the 18S rRNA gene is widely recognized as a hypervariable region, to the extent that it is used as a barcode in environmental studies. Therefore, significant variations in the V4 region secondary structure are to be expected.

In my opinion, the primary limitation of this study is the exclusive use of a single species as outgroup in the analyses. Given the highly variable nature of the selected marker, it is not possible to make an absolute assertion that the ancestral character is identical to that observed in Certesia quadrinucleata. The authors themselves found that the secondary structure types I and II appear several times independently in the different Euplotes lineages, both in clades considered "basal" and "more recent." It is therefore recommended that the authors redo the analysis, including at least five species as outgroup, as other spirotrichs and even ciliates from other classes.

Experimental design

Molecular clock analysis is lacking in adequacy. While the authors based on Wright and Lynn's pioneering work (1997), recent studies offer a more comprehensive dataset for calibrating the tree, including the timing of origin and diversification of spirotrichs like Euplotes. These studies could have been used as a basis for calibrating the authors' tree. Notably, the estimate in Figure S3 places the origin of Euplotes between 700 and 1200 million years ago (mya), which contradicts recent findings placing it around 650 mya, during the transition between the Cryogenian and Ediacaran periods (Fernandes and Schrago, 2019). Furthermore, the low precision of the estimates, due to the highly variable marker used, also contributes to this discrepancy. This analysis doesn't need to show that clade I includes the earliest species to diverge from the common ancestor of Euplotes spp., or that clade VI contains the most recently evolved. For this study, molecular clock analysis isn't crucial. Adding more representatives to the outgroup in the phylogenetic analysis would be more meaningful. The molecular clock analysis should be redone with proper references and calibrations, or removed from the paper. This will not affect the conclusions reached.

Validity of the findings

In the conclusion, the authors posit that: “In essence, this study not only sheds light on the existence of cryptic species within Euplotes but also highlights the potential for reassessing the taxonomic determination of certain species based on molecular and reproductive compatibility.” - The debate over cryptic species in Euplotes has been long-standing (Valbonesi et al. 1992, Petroni et al. 2002, Zhao et al. 2018, to name just a few). Furthermore, the use of molecular data in phylogenetic reconstructions as a basis for reorganizing the taxonomy of practically all biological groups is also a well-established practice. This sentence prompts me to question the real relevance of this study.

Additional comments

Minor but not less relevant issues:

Line 27: This section requires further detail. How many species were analyzed?

Line 29: “Ancestral state reconstruction indicated a transition from a simpler type (type I) to a more complex type (type II) secondary structure, with certain groups showing a reversal to type I” - Here, with the expression "certain groups", the authors also refer to the more derived clades (clade VI), in which the characteristic judged to be "primitive" reappears, which is inconsistent with the following sentence: “Notably, basal groups exhibited more primitive characteristics, both morphologically and molecularly, compared to later divergent groups”.

Line 32: "molecular clock analysis" - As the authors utilized molecular clock analyses for these statements, such analyses should have been more thoroughly discussed in the text.

Figure 1: Which prokaryotes are the authors referring to? It's important to note that prokaryotes are paraphyletic. Are these helices also observed in archaea or bacteria?

Line 147: The authors incorrectly employ evolutionary concepts, as seen in the sentence: "indicating addition of two helices is more advanced than type I". The terms "more advanced", "more primitive", "in more evolved species", appear several times in the text, which is inappropriate for evolutionary studies.

Line 154: “Clade VI seems to represent a relatively recent clade in the genus Euplotes phylogenetic tree, and these results suggest a notable occurrence of more primitive characters reappearing in the most advanced species within this clade” - Using only one species as an outgroup it cannot be affirmed.

Line 160: Authors should specify which definition of Reverse evolution they are using and the reference.

Line 164-166: This statement contradicts the results, as Type I is also observed in derived lineages.

Line 177-180: Nothing mentioned in this section is novel.

Line 199-202: “both molecular evidence from V4 secondary structure and morphological features collectively support the notion that species in the basal group (Clade I) of genus Euplotes are characterized by more primitive features compared to species in later divergent groups (Clade II to VI) (Syberg-Olsen et al., 2016; Zhao et al., 2018)”- Apart from the analysis of the V4 secondary structure, the study does not introduce any novel findings.

Line 204: “The difference in secondary structure of V4 reveals cryptic species within genus Euplotes.” - Once again, the revelation of cryptic species in Euplotes is not primarily attributed to the secondary structure of V4, as this has been previously investigated. I suggest rephrasing the sentence to emphasize the genuine contribution of this study.

Reviewer 2 ·

Basic reporting

This work focuses on the model organism Euplotes in unicellular eukaryotes. Through the secondary structure prediction and CBC analysis of the V4 region of ribosomal genes, it aims to reveal the internal evolutionary process of Euplotes and uncover hidden species. The author found that there are obvious differences in the V4 region among different clades within the genus Euplotes. The secondary structure of the V4 region becomes more complex from simple in the evolutionary process of the genus, which is consistent with the evolutionary pattern of morphology. The author also found that CBC analysis can distinguish several species of Euplotes.

The result is technically sound and the major conclusions of the paper are supported. But I also think the manuscript needs to be improved by detailing the Materials and Methods, elaborating figures and illustrations, and formatting the literature.

I list some major points here:
1. The reason for selection of the analysis method CBC and the SSU variable 4 region is insufficiently introduced. The author mentioned in the results that CBC was used to distinguish species. However, there is a lack of introduction to the validation and advantage of CBC either in the introduction or in the materials and methods. The author chose the V4 region to predict the secondary structure, but the intention for choosing this region was not clarified. Why other variable regions was not selected, such as the V2 region and the V9 region?

2 The illustrations in the manuscript need to be more refined and detailed. The author mentioned many features of the secondary structures in the results, and these features need to be clearly presented using diagrams and labels in the main text. For example, in lines 123-129, when referring to hairpin loop, long or short helix E23_8, additional helices, etc., it is relatively obscure to only use language description, and it will be more clear and easier to understand by using illustrations. I would recommend to give a figure in the main text, to show secondary structures of representatives for each of the six clades. In the figure, the distinct features described above should be labeled in detail with arrows, color blocks, words, etc. One literature can be referred: Zhang Q, Yi Z, Fan X, Warren A, Gong J, Song W. 2014. Further insights into the phylogeny of two ciliate classes Nassophorea and Prostomatea (Protista, Ciliophora). Mol. Phylogenet. Evol. 70: 162-170.
In terms of the phylogenetic tree, basic information such as the support value and substitute rate is lacking in the figure, and the sequence source based on which the phylogenetic tree is built, and the analysis method used and other basic information are not given in the figure caption. The font of the species name in Figure 2 is so small that it is almost indistinguishable.

3 The materials and methods of the manuscript is too simple. The sample source and the sequence acquisition process are completely lacking. The basis for the selection of the V4 region is not given. The specific analysis method of CBC is missing. The method to calculate the molecular clock is lacking. The construction of the phylogenetic tree lacks specific information.

4 There are a large number of formatting errors in the literature part of the manuscript. The most prominent point is that the magazine names are not unified, with full names and abbreviations being used interchangeably, and the first letters of the magazine names are not all capitalized.

Experimental design

The reason for selection of the analysis method CBC and the SSU variable 4 region is insufficiently introduced. The author mentioned in the results that CBC was used to distinguish species. However, there is a lack of introduction to the validation and advantage of CBC either in the introduction or in the materials and methods. The author chose the V4 region to predict the secondary structure, but the intention for choosing this region was not clarified. Why other variable regions was not selected, such as the V2 region and the V9 region?

Validity of the findings

no comment

Additional comments

Detailed comments:
Line 68-70: Are there any studies of rRNA secondary structure on the genus Euplotes?
Line 71: CBC needs to be introduced here, about its validation on species distinguish and supportive literatures.
Line 79: The acquisition method of the SSU rRNA sequence is lacking. What method was used to extract the DNA and amplify SSU gene sequences? Did you apply single cell or cultured multiple cells? Is it through cloning method or direct sequencing by PCR? Is the sequence method bidirectional sequencing or unidirectional sequencing, etc…
Line 81: The origin of the samples and the basis for species identification needs to be specified. Were these specimens collected from cultures or natural environment? How did you identify species accurately? Are there microscopy or silver staining data?
Line 104: The specific analysis methods of the CBC is lacking.
Line 105-107: The specific steps of phylogenetic tree construction need to be given. For example, based on which segment of the SSU rRNA sequence the tree was constructed? How did you choose the evolutionary model? How did you set the number of bootstraps, etc. In addition, there is no support values shown on the phylogenetic tree in Fig. 3, and it is not known whether the homology of each clade is supported.
Line 116-117: These characters need to be noted in the figure using arrows or other symbols.
Line 123-129: Please give another formal and elaborated figure to show these characters and clearly note the differences.
Line 137-138: The validation and reason for application of CBC in determining genetic differences need to be clarified in somewhere before this section.
Line 142-144: Which figure or table is the result displayed?

---

## Round 0.2 · Minor Revisions

Dear Dr. Wardani and colleagues:

Thanks for revising your manuscript. The reviewers are mostly satisfied with your revision (as am I). Great! However, there are a few concerns to address per reviewer 2. Please attend to these issues ASAP so we may move towards acceptance of your work.

Best,

-joe

Reviewer 2 ·

Basic reporting

The manuscript has supplemented the background part and materials and methods, and also improved the figure of the secondary results. Otherwise, in my opinion, the current version can still be improved by addressing the following problems:
1. The article contains a large number of spelling errors such as missing spaces, commas and repeated sentences. This may be the consequence of not checking the whole article after removing the revision format.
2. In the results section, a link to the specific subfigures (e.g. Figure 1A, or Figure S1B) should be given after every 1-2 sentences, to facilitate readers to understand the author's description of the text. However, in almost all paragraphs in the results, only one figure link was given at the very end.
3. I think supplementary figure 2 should be moved to the main text as formal figure. As can be seen, significant portion of the Result section 1 is describing the Fig S2, but we can only refer supplementary files for illustrations. In addition, "type I" and "type II" should be labeled in Fig S2.
4. In Figure 3, perhaps two simple schematic diagrams of secondary structures can be placed in the legend to show the typical characteristics of "type I" and "type II". In the current version, there is only text, and readers can only go back to Figure 1 to review the structural characteristics of "type I" and "type II".

Other minus comments:
Line 121-122: There should be specific data, rather than general description for this feature. For example, how much longer, how many mutations more?
Line 124-125: The latter half of the sentence is incomprehensible.
Line 129: typing error
Line 167: For example, here should be some link to the figure S2 after the “…. helix E23_8”.
Line 183: The coined name “ vannus-crassus complex” is not proper. The genera name Euplotes should be added. In addition, how do you define the name of this complex? It seems it comprises at least 6 species.
Line 188: “The ancestral state”: How did you analyze this is not clearly clarified?
Line 192: “…..retained in members of clade I and II”: Where can we find this result from figures? The same question for the following sentences.
Line 210: “…….observed in Euplotes members”, and the outgroups of genus Euplotes?
Line 213-214: I don't understand. Does this sentence mean that the conclusion of this paragraph is derived from phylogenetic analysis and molecular clock? Then what is the relationship between this and the content of the secondary structure mentioned above?

Experimental design

no comments

Validity of the findings

no comments

Additional comments

no comments

---

## Round 0.3 · accepted · Accept

Dear Dr. Wardani and colleagues:

Thanks for revising your manuscript based on the concerns raised by the reviewer. I now believe that your manuscript is suitable for publication. Congratulations! I look forward to seeing this work in print, and I anticipate it being an important resource for groups studying Euplotes evolution and speciation. Thanks again for choosing PeerJ to publish such important work.

Best,

-joe